# Evaluating the use of social contact data to produce age-specific short-term forecasts of SARS-CoV-2 incidence in England

**James D. Munday**[1,2,3]*, **Sam Abbott**[1,2], **Sophie Meakin**[1,2], **Sebastian Funk**[1,2]

**1** Centre for Mathematical Modelling of Infectious Diseases, London School of Hygiene and Tropical Medicine, London, United Kingdom, **2** Department of Infectious Disease Epidemiology, London School of Hygiene and Tropical Medicine, London, United Kingdom, **3** Department of Biosystems Science and Engineering, ETH Zürich, Basel, Switzerland

* james.munday@bsse.ethz.ch

## Abstract

Mathematical and statistical models can be used to make predictions of how epidemics may progress in the near future and form a central part of outbreak mitigation and control. Renewal equation based models allow inference of epidemiological parameters from historical data and forecast future epidemic dynamics without requiring complex mechanistic assumptions. However, these models typically ignore interaction between age groups, partly due to challenges in parameterising a time varying interaction matrix. Social contact data collected regularly during the COVID-19 epidemic provide a means to inform interaction between age groups in real-time. We developed an age-specific forecasting framework and applied it to two age-stratified time-series: incidence of SARS-CoV-2 infection, estimated from a national infection and antibody prevalence survey; and, reported cases according to the UK national COVID-19 dashboard. Jointly fitting our model to social contact data from the CoMix study, we inferred a time-varying next generation matrix which we used to project infections and cases in the four weeks following each of 29 forecast dates between October 2020 and November 2021. We evaluated the forecasts using proper scoring rules and compared performance with three other models with alternative data and specifications alongside two naive baseline models. Overall, incorporating age interaction improved forecasts of infections and the CoMix-data-informed model was the best performing model at time horizons between two and four weeks. However, this was not true when forecasting cases. We found that age group interaction was most important for predicting cases in children and older adults. The contact-data-informed models performed best during the winter months of 2020–2021, but performed comparatively poorly in other periods. We highlight challenges regarding the incorporation of contact data in forecasting and offer proposals as to how to extend and adapt our approach, which may lead to more successful forecasts in future.

**Data Availability Statement:** Data availability statement Case data is available on the UK Covid-19 Dashboard https://coronavirus.data.gov.uk/details/cases?areaType=nation%26areaName=

England#card-cases_by_specimen_date Infection and antibody data are available on the Covid-19 infection survey website https://www.ons.gov.uk/peoplepopulationandcommunity/healthandsocialcare/conditionsanddiseases/bulletins/coronaviruscovid19infectionsurveypilot/25november2022 and contact matrices are available from the CoMix online repository https://doi.org/10.5281/zenodo.7351951. Code availability statement All code used for this work is available at https://github.com/epiforecasts/CovidAgeGroupForecast.

**Funding:** JDM's work was supported by Office for National Statistics COVID-19 Infection Survey Analysis grant PU-20-0205(c) and by the Wellcome Trust (grant: 210758/Z/18/Z, https://wellcome.org/). SA's work was supported by the Wellcome Trust (grant: 210758/Z/18/Z, https://wellcome.org/). SM's work was supported by the Wellcome Trust (grant: 210758/Z/18/Z, https://wellcome.org/). SF's work was supported by the Wellcome Trust (grant: 210758/Z/18/Z, https://wellcome.org/). The funders had no role in study design, data collection and analysis, decision to publish, or preparation of the manuscript.

**Competing interests:** The authors have declared that no competing interests exist.

## Author summary

Short term epidemic forecasts help policy makers to plan and implement response activities. It can be useful to have such forecasts separately for different age groups, in order to reflect potential differences in transmission and incidence of infection between age groups as well as differences in risk of severe disease or death. A key challenge in developing age-specific models is understanding how different age-groups interact. We used data collected during a large-scale weekly survey of social contacts in the UK to inform this interaction in a model for short-term forecasts of COVID-19. To assess whether allowing interaction between age-groups and the use of contact data improved forecasts, we compared our forecasts to those from a set of models that either didn't use current contact data or treated each age group as a separate population. We found that including timely contact data improved predictions when forecasting two to four weeks into the future, but this improvement was not consistent throughout the epidemic. The best improvement was measured during a long national "lockdown". We also found that inclusion of age-group interaction and use of contact data were most important when forecasting infections in older adults and young children.

## Introduction

Effective epidemic response relies on accurate infection surveillance to provide status updates which support decision makers [1]. Surveillance data can be enhanced by estimating key epidemiological parameters in real-time such as the growth rate and time-varying reproduction number ($R_t$) and by generating short-term forecasts of incidence of infection, hospitalisation and mortality [2–4]. These provide estimates of the current and future epidemic trajectory to public health decision makers. As such a host of approaches have been developed to make short term epidemiological forecasts. A popular genre of methodology for infectious disease forecasts are renewal equation based 'semi-mechanistic' models [2,4–6], which infer key epidemiological parameters from historical time-series data, in particular changes in transmission intensity, and use them to forecast future epidemic dynamics without requiring the more detailed assumptions and complex mathematical framework involved in 'fully-mechanistic' models (e.g. compartmental or agent based models).

Age has been shown to be an important factor in both transmission risk [7,8] and severity of disease [9–11] caused by SARS-CoV-2. This is not unique to the COVID epidemic. In the past, epidemiological analysis and modelling have shown that variability and homophily in transmission by age have important implications for the dynamics of infection [12–15]. Moreover, age distribution of infection has important implications for the potential burden of disease as infection moves between age groups, who are more and less prone to severe illness and death [7,8,16].

Throughout the pandemic there has been a great deal of variance in prevalence of infection between age groups. Although this further motivates the requirement for age-specific forecasts, especially to better understand the risk to particularly vulnerable groups, this heterogeneity in prevalence also suggests that transmission rates between age groups may need to be captured to effectively make such forecasts. However, the high dimensionality of this problem means that the full age interaction matrix is not identifiable from epidemiological data alone [17]. Instead, much infectious disease dynamics research in the past 30 years has made assumptions in line with the social contact hypothesis [17]. It states that the rate of transmission of directly infectious agents is proportional to the population-level rate of social contact between

population groups. This hypothesis is the basis for age-structured mixing assumptions in many mathematical models. Such models are generally parameterised from data gathered in social contact surveys [15,18], which typically ask participants to report their social contacts from a fixed period in the recent past, e.g. the last 24 hours. Participants are also asked about the characteristics of their contacts at each contact event, usually including age [15]. A key challenge to the use of historically collected contact data has been of their temporal and geographical generalisability. This is especially true when non-pharmaceutical interventions (NPIs) are in effect, potentially drastically changing the contact behaviour of the general public. The variability in behaviour with time and age during a pandemic makes parameterisation of age-specific real-time models particularly challenging as up-to-date information on interaction is essential for time-varying parameterisation of the model.

During the COVID-19 pandemic, as a means to monitor the behaviour of the general public relevant to transmission and provide insight into risk posed to vulnerable populations, a number of studies were conducted to survey social contacts at a frequency and scale not seen previously. One example is the CoMix study, which collected contact data weekly between March 2020 to March 2022 in 19 European countries [19–22]. The UK arm of the study, which involved a survey of greater than 5000 participants, was the first to launch and most complete in terms of data collected over this period. This regularly collected contact data provides a means to parameterise models with temporally and geographically relevant estimates of social interaction, and an opportunity to evaluate how incorporating such data into a real-time analysis framework impacts forecast performance at different scales.

Existing studies of forecasting performance [5,6,23] have focused on age-agnostic numbers of cases, hospitalisations and deaths. Probabilistic forecasts can be robustly assessed using proper scoring rules [24]. Although these methods have been popular for some time in other fields, they have only recently been applied to epidemic forecasts [5,6,23]. To the authors knowledge one such evaluation has previously been made [25] of age-stratified epidemic forecasts however, the study by Held et. al. used historical contact data to parameterize interaction between age groups and evaluated at a population level by summing age-specific forecasts. To our knowledge there has been no evaluation of the use of the regularly collected age-stratified contact data in comparison to other approaches to make short term forecasts at an age group specific level.

Here we present age-specific forecasts in the UK, with the aim of understanding whether incorporating the weekly collected social contact data improves the predictive ability compared to ignoring this interaction. We incorporated data from the CoMix study in a semi-mechanistic forecasting framework and applied this to case numbers, as the most commonly tracked metric for COVID-19 dynamics in the UK throughout the pandemic. We further applied it to infection incidence estimated from a weekly cross-sectional household survey of infection [26,27] in order to better understand the influence of reporting patterns on results. To quantify the relative benefits of incorporating interaction between age groups and specific contact data into forecasts we compared three models with interaction between age groups with an equivalent model with no such interaction and evaluated the models against two naive baseline models.

## Materials and methods

### Study overview

To establish the relative benefit of incorporating interaction between age groups in short-term epidemiological forecasts, we implemented four age-stratified semi-mechanistic models, which each estimate a time-varying Next Generation Matrix (NGM). This matrix is inferred as

the interaction matrix between age groups under the assumption that all infections in each age group are informed by the sum of past infections in all age groups weighted by the distribution of time between infections, the generation interval distribution and the NGM. Two of the models included interaction between age groups: The first of which was informed by contact data from the CoMix study (regularly collected during the period of study). To evaluate the benefit of regularly collected contact data over interactions informed by previously collected contact data, we implemented a second instance of the same model using data collected in the POLYMOD survey. This survey conducted in 2008 was designed to measure rates of contact in the population for parameterisation of mathematical models of infectious transmission. The POLYMOD data represent contact during a non-pandemic period and in our study represented an interaction assumption that may be made if the regularly collected CoMix data were not available. Gimma et. al. [19] compared CoMix survey rounds and POLYMOD data in detail, showing that the mean contact rates varied substantially in all age groups between CoMix and POLYMOD, but also between the rounds of CoMix. In particular, the relative rate of contact in children varied between rounds. For example, compared to POLYMOD, there was a much greater reduction in childrens' contacts than adults' during periods that schools were closed, however as schools re-opened the childrens' contacts almost returned to parity with POLYMOD, whereas adults' contacts remained reduced. Hence the CoMix data added information on both time-varying mean contacts and variation in the interaction between age-groups.

In the third model the interaction between age groups was not informed by contact data at all, but estimated entirely from historical epidemiological data. We do not anticipate the inferred contact matrix to necessarily reflect the true contact rates due to the aforementioned challenges with identifiability. Rather, we wish to test if an over-fit contact matrix from this model performs better than the model based on the collected data. We compared these models with a fourth model which allowed no interaction between age groups.

We applied this to reported cases, as a commonly available quantity for forecasting epidemic dynamics [5]. This, however, incurs a secondary challenge due to potential variability in reporting of cases by age and over the course of an epidemic, which may serve to complicate our interpretation of the application of contact data to forecasts. Hence, to isolate the impact of incorporating contact data we chose to additionally apply the models to estimated infection incidence from a repeated cross-sectional household survey of infections.

We made forecasts of weekly reported cases using the data from the UKHSA Covid-19 dashboard. For convenience we used the full case time-series aggregated to weekly incidence and truncated at different forecast dates, rather than the data available on each forecast date. Although this does not give a full picture of the real-time applicability and performance of the model, it avoids complications in delays in gathering case reports which require additional treatment prior to application of a forecasting model such as truncation of the most recent data or now-casting [28]. Secondly, we applied the models to estimates of weekly incidence of infection estimated [26] from national infection prevalence data, again with the full final data set truncated at each forecast date rather than snapshots available at the time. To further isolate the role of contact data in the forecasts of infections, we used weekly age-stratified estimates of antibody prevalence to inform age-specific susceptibility.

## Data

We accessed daily, age-stratified, case data from the UK COVID-19 dashboard [29] on 11th May 2022. We aggregated this data to weekly incidence by taking the sum of the previous 7 days, aligned such that the weekly data is reported on the proposed forecast dates, to forecast

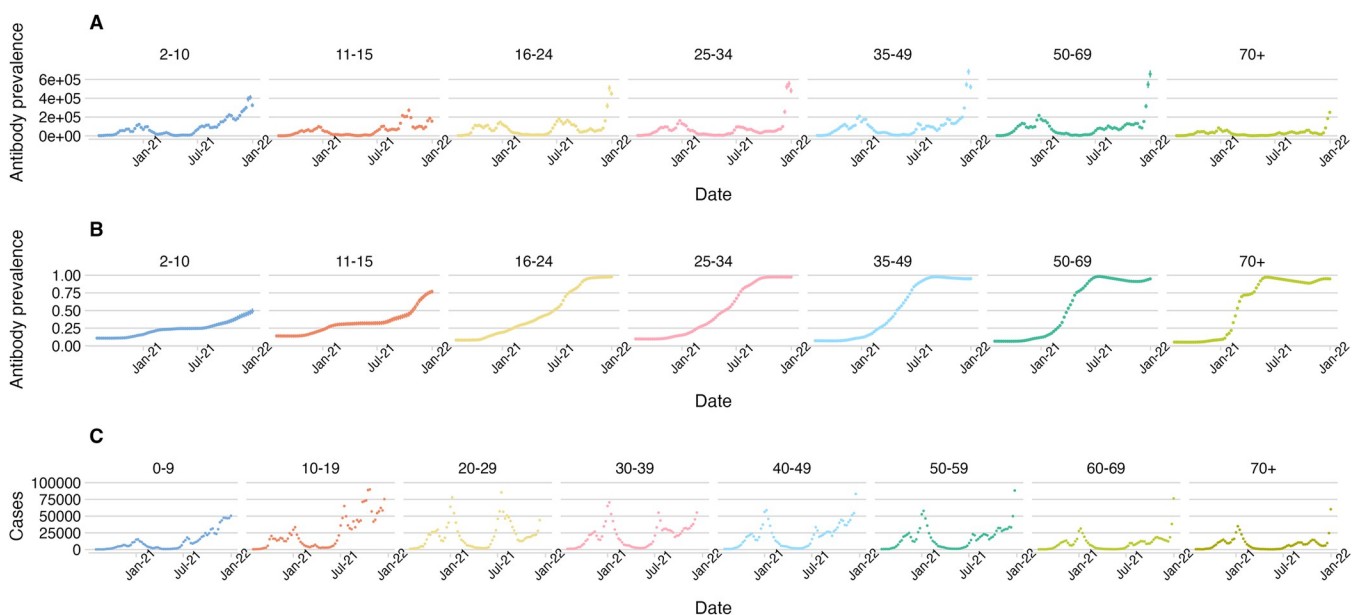

**Fig 1.** Estimated incidence of A) infection, B) antibody prevalence from the ONS Community Infection Survey (CIS) and C) case reports from the UKHSA COVID-19 Dashboard, each shown by age group.

weekly case counts in future weeks. The case reports were grouped in seven decade groups between zero and 69 years with a single group for over 70 year olds (0–9, 10–19, . . ., 60–69, 70 +). Dates in the case data corresponded to the day on which the infected individual was swabbed.

We accessed aggregates of SARS-CoV-2 infection prevalence and antibody prevalence collected as part of the COVID-19 infection survey (CIS) through the CIS Website [27] on 18th March 2022. Here infection prevalence is defined as PCR test positivity prevalence under the protocol of the CIS. We used data covering the period between August 2020 and January 2022 to estimate weekly infection incidence and antibody prevalence for seven age groups (2–10, 11–15, 16–24, 25–34, 35–49, 49–69 and 70+). In addition to the CIS data, we used vaccination data published by the National Health Service and accessed via the UK coronavirus dashboard [29] on the same date (Fig A in S1 Text).

We generated SARS-CoV-2 infection incidence and antibody prevalence time-series for the period between August 2020 and January 2022 using an approach described elsewhere [26,30] (Fig 1), henceforth described as the inc2prev model. To establish a weekly time-series of infections we took the sum of incident infections in each week on a sample-by-sample basis and calculated the credible intervals from the resultant sum, this constitutes the distribution of weekly infections according to our modelled estimates. To establish a weekly time series of antibodies we took the antibody prevalence distribution on the last day of each week and calculated credible intervals from the full posterior sample. We accounted for the uncertainty in the estimated antibody prevalence and infection incidence by including them as latent parameters in our model with the prior at each day informed by the estimated distributions above (see the Transmission Model section of this paper for more details)

We combined these data with social contact data collected as part of the CoMix social contact survey (CoMix) [19,31], a multinational, weekly, cross-sectional survey of social contacts. We used published weekly contact matrices from the UK arm of CoMix, generated under the framework described previously [20].

## Transmission model

We extended the concept of the Next Generation Matrix to include transmission interval distributions (the generation interval for infections, and the interval between a positive test in infector and infectee for cases). Here, the number of incident cases or infections I($t$) at time $t$ was given by the sum of the products of the next generation matrix **N** and the age-stratified vector of cases or infections on dates between $t$-$s_{max}$ and $t$-1, weighted by the transmission interval distribution $w(s)$.

$$\overrightarrow{I}(t) = \sum_{s=1}^{s_{max}} w(s) \times N(t-s)\overrightarrow{I}(t-s) \tag{1}$$

where $s_{max}$ is a fixed upper-limit of the transmission interval distribution, set to 4 weeks and $w(s)$ is assumed to follow a discretised log-normal distribution with time since the primary event (infection or positive test of the infector):

$$w(s) = (F_{LNorm}(s, w_\mu, w_\sigma) - F_{LNorm}(s-1, w_\mu, w_\sigma))/F_{LNorm}(s_{max}, w_\mu, w_\sigma) \tag{2}$$

where $F_{LNorm}$ is the cumulative distribution function of the log-normal distribution with parameters $w_\mu$ and $w_\sigma$. Under the social contact hypothesis [17], the next generation matrix is calculated by multiplying the contact matrix, **C**($t$) quantifying the mean number of contacts between age groups, with vectors of age-specific susceptibility, $\overrightarrow{s}$, and infectiousness, $\overrightarrow{i}$, where each element, $s_a$ and $i_a$ give the specific susceptibility and infectiousness of age group, $a$ [13].

$$N(t) = \text{diag}(\overrightarrow{s})C(t)\text{diag}(\overrightarrow{i}) \tag{3}$$

We assumed that age specific infectiousness, is inherent and unrelated to time varying factors associated with the epidemic. We assumed that age-specific susceptibility included two components:

$$s_a = s_{ab,a}s_{inh,a} \tag{4}$$

The first ($s_{ab,a}$) is the age-specific acquired immunity to infection in age group a, this is informed by weekly antibody prevalence estimates from inc2prev. We used a leaky definition of antibody effectiveness in line with the definition used in the estimation of the infection and antibody timeseries:

$$s_{ab,a} = 1 + (\Phi - 1)A_a(t) \tag{5}$$

Where $\Phi$ is the effectiveness of antibodies in preventing infections in an exposed member of the population and $A_a(t)$ is the antibody prevalence in age-group $a$ at time $t$. The second component ($\overrightarrow{s}_{inh}$) is due to an age-correlated variation in inherent susceptibility to infection and unrelated to time-varying factors associated with the epidemic. Both $\overrightarrow{s}_{inh}$ and $\overrightarrow{i}$ were assumed to remain constant in time, such that all of the variation in the next generation matrix by time is governed by changes in contacts and estimated antibody derived immunity. Both $\overrightarrow{s}_{inh}$ and $\overrightarrow{i}$ were fit as random effects in a hierarchical framework (Table 1). To account for uncertainty in estimates of antibody prevalence, $A_a(t)$ was also estimated in our model, the prior was based on the mean and standard deviation of the previously estimated antibody prevalence using inc2prev ($A_{\mu,a}(t)$, $A_{\sigma,a}(t)$).

**Table 1. Model parameters and priors.**

| parameter | Symbol | Prior | Ref. |
|---|---|---|---|
| Antibodies | $A_a(t)$ | $A(t) \sim \mathrm{normal}(A_{a,\mu}(t), A_{a,\sigma}(t))T[0,1]$ | |
| Antibody protection | $\Phi$ | $\phi \sim \mathrm{gamma}(2,2)T[0,1]$ | |
| Generation interval distribution log-mean and log-variance | $w(t)$ | $w_{mean} = 5/7$ $w_{sd} = 5/7$ $w_{\sigma.prior} = \log(((w_{sd}^2)/(w_{mean}^2))+1)$ $w_{mu.prior} = \log(w_{mean}) - (w_{\sigma.prior})/2$ $w_{mu} \sim \mathrm{normal}(w_{\mu.prior}, w_{\mu.prior}/5)T[0,]$ $w_{\sigma} \sim \mathrm{normal}(w_{\sigma.prior}, w_{\sigma.prior}/5)T[0,]$ | [34–36] |
| inherent susceptibility | $s_{inh}$ | $s_{mu}^h \sim \mathrm{Beta}(24,24)$ $s_{sd}^h \sim \mathrm{normal}(0.1,0.02)T[0,]$ $s_{inh,a}' \sim \mathrm{normal}(0,1)$ $s_{inh,a} = s_{mu}^h + s_{sd}^h s_{inh,a}'$ | [16] |
| inherent infectiousness | $i_a$ | $i_{mu}^h \sim \mathrm{Beta}(12,4)$ $i_{sd}^h \sim \mathrm{normal}(0.1,0.02)T[0,]$ $i_a' \sim \mathrm{normal}(0,1)$ $i_a = i_{mu}^h + i_{sd}^h i_a'$ | [16] |
| Contact matrices | C | $C_{ab} \sim \mathrm{gamma}(2,2)$ | |
| Uncertainty in infections | $CV_I$ | $CV_I \sim \mathrm{normal}(0.05,0.025)T[0,]$ | |
| Uncertainty in cases | $CV_c$ | $CV_c \sim \mathrm{normal}(0.05,0.01)T[0,]$ | |
| Uncertainty in contacts | $\sigma_{cm}$ | $\sigma_{cm} \sim \mathrm{normal}(0.05,0.025)T[0,]$ | |

Where $T[a,b]$ indicates distribution is truncated between the values $a$ and $b$

## Parameter estimation and forecasting

To allow variation in parameter values over the course of the study period, we parameterised the model with the estimated antibody prevalence and contact matrices and fitted it to 8 weeks of weekly estimated infection or reported case data prior to the forecast date. We fitted the model using Hamiltonian Monte Carlo, implemented in the Stan probabilistic programming language [32] we assessed the performance of the model fits by monitoring convergence metrics and transition divergences.

We fitted to the mean infection time series under the likelihood:

$$\overrightarrow{I_\mu}(t) \sim \mathrm{normal}(\sum_{s=1}^{s_{max}} w(s)N(t-s)\overrightarrow{I_\mu}(t-s), \sigma\prime_I(t)) \qquad (6)$$

Where $\overrightarrow{\sigma\prime_I}(t)$ is the vector of the overall uncertainty in the modelled infections for each age group., Each age-group element (e.g. σ'$_{I,a}$, for age group $a$) combines the time-varying, age specific inherent uncertainty in the NGM model, $\sigma_{I,a}$, and the standard deviation of the infection estimates, $I_{\sigma,a}$,

$$\sigma_{I,a}'(t) = \sqrt{I_{\sigma,a}^2 + \sigma_{I,a}(t)^2} \qquad (7)$$

$\sigma_{I,a}$ is constructed at each time point from the estimated coefficient of variation $CV_{I,a}$ and infection incidence such that:

$$\sigma_{I,a}(t) = CV_{I,a}I_{\mu,a}(t) \tag{8}$$

Where $I_{\mu,a}(t)$ is the mean of the infections estimated using *inc2prev* in age group a. This ensures the uncertainty scales with the magnitude of the infection incidence estimates. We fit to the case time series $c(t)$ under the likelihood:

$$\overrightarrow{c}(t) \sim \text{normal}(\sum_{s=1}^{s_{max}} w(s) \times N(t-s)\overrightarrow{c}(t-s), \overrightarrow{\sigma_c}(t)) \tag{9}$$

Where $\sigma_C$ is the modelled uncertainty in cases and is constructed from the estimated coefficient of variation $CV_C$ at each time point for each age-group ($a$) such that:

$$\sigma_{c,a}(t) = CV_{c,a}c_{\mu,a}(t) \tag{10}$$

which ensures the uncertainty scales with the magnitude of the reported case incidence. To incorporate the contact data in the CoMix based model we jointly fit the contact matrices under the likelihood:

$$C_{ab}(t) \sim \text{normal}(C_{\mu,ab}, \sigma'_{cm,ab}) \tag{11}$$

Where, $C_{ab}(t)$ is the rate of contact between age groups a and b at time t, $C_{\mu,ab}$ is the mean rate of contact recorded between age groups $a$ and $b$, and

$$\sigma'_{cm,ab} = \sqrt{C^2_{\sigma,ab} + \sigma^2_{cm}} \tag{12}$$

and $C_{\sigma,ab}$ is the observed standard deviation of the measured contact rate and $\sigma_{cm}$ is the uncertainty in the posterior contact rates.

Each of the models estimated a NGM which varied over the 8 weeks of prior data only by changes in the estimated contact matrices and antibody inferred immunity, whilst the inherent susceptibility and infectiousness vectors were assumed constant for the whole modelled period. However, as each forecast date was modelled independently, all parameters were able to vary between forecasts.

The priors we employed are given in Table 1. Antibody prevalence priors were set to the distribution of the estimate provided by the model used to estimate incidence [30] and relative susceptibility and infectiousness vector elements were set such that the Secondary Attack Rate (SAR) was roughly half that of estimates of Household SAR in literature [33]—which aimed to account for reduced risk of transmission to known contacts outside the household. The prior for the log-mean ($w_\mu$) and log-standard-deviation ($w_\sigma$) of the transmission interval had a mean and standard deviation of 5 days to reflect the broad distribution or transmission intervals recorded in literature [34–36], these were converted to the appropriate log-parameters for the log-normal framework in Eq 2, and their prior was set to be normally distributed with a standard deviation of 20% of the mean.

We used posterior distributions of the parameters (Table 1) to project infections and cases forwards up to four weeks after each forecast date. We note that contact data directly relevant to the dates forecasted would not be known on the forecast date, so we used the contact data corresponding to the week of the forecast date itself, assuming that these also reflected contacts in the following week. For the case forecasts we offset the contact data by 7 days to account for delay between infection and specimen date and used the generation interval as a proxy of the

test-to-test distribution [37], which is consistent with a 5 day incubation period and a 2 day report delay [38].

## Model evaluation

We evaluated the performance of the NGM models (CoMix-data, POLYMOD-data, No-contact-data and No-interaction) across 29 forecast dates between October 2020 and December 2021. We chose this period as there was major disruption to the CoMix survey during July 2020 and following changes in the survey in June 2020. We excluded dates after December 2021 due to the complication of the emergence of the Omicron variant, which has been shown to evade immunity to a greater extent than earlier variants [39]. This would complicate our interpretation of antibody prevalence as a mix of Omicron-specific and previously acquired antibodies persist in the population.

We evaluated the forecasts against the reported number of cases or mean estimated number of infections in the week forecasted, for case and infection forecasts respectively. We evaluated the forecasts based on Continuous Ranked Probability Score (CRPS) and a measure of bias (see appendix for definitions) each implemented using the *scoringutils* R package [40]. The CRPS measures the 'distance' of the predictive distribution to the observed data-generating distribution, hence a lower score indicates more accurate predictions and therefore a higher performing model. To summarise the performance of the entire model, we calculated the overall CRPS as the mean CRPS across all age-groups. The bias measures the tendency for a model to over (positive value) or under (negative value) predict the incidence in its projections, hence a bias of zero is optimal.

We also assessed the models calibration by evaluating the central interval coverage, henceforth referred to as coverage, of each model's forecast (Proportion of incidence points which fell in the ranges projected by the forecast model's posterior distribution of future cases).

To provide a comparator as a lower bound of performance, we also evaluated two baseline models. These baselines were intended to represent naive assumptions, which may be applied without the use of a model. The first baseline assumed no change in incidence from the day the forecast was made. The second calculated the change in incidence between the forecast date and each week within the four week forecast horizon, the rate of change is projected as an exponential extrapolation based on the previous two weeks of data. Both baselines were modelled without uncertainty and, consequently, the CRPS reduced to the mean absolute error. To provide a clear comparison of performance with and without interaction between age groups, we provide all CRPS scores relative to the score of the no-interaction model (rCRPS).

As well as the overall performance of each forecasting model, we also evaluated the forecasts by grouping forecasts made by each model in two ways. Firstly, we aggregated the forecasts by age group—showing the relative performance of the models when forecasting incidence in particular age categories. Secondly, to evaluate how performance changed over the course of the analysis period, we scored the forecasts separately for seven key periods of the pandemic (Table 2). For this we used the periods used in Gimma, et. al. [19] which overlapped with our

**Table 2. Pandemic period names and dates.**

| Period | Start date | End date | Characteristics |
|---|---|---|---|
| Lockdown 2 | 2020-11-05 | 2020-12-02 | Increasing |
| Lockdown 2 Easing | 2020-12-03 | 2020-12-19 | Increasing |
| Christmas | 2020-12-20 | 2021-01-04 | Decreasing |
| Lockdown 3 | 2021-05-01 | 2021-03-08 | |
| Lockdown 3 Schools open | 2021-03-09 | 2021-03-28 | Decreasing to increasing change |
| Lockdown 3 easing | 2021-03-29 | 2021-09-30 | |
| Opening up | 2021-10-01 | 2021-11-24 | Stable |

analysis, with the addition of two periods that were not covered by the previous CoMix work. Due to the small number of weeks covered by 'Christmas' and 'Lockdown 3 schools open' we combined these with 'Lockdown 3' and 'Lockdown 3 Easing' respectively. The periods we analysed are defined by the non-pharmaceutical interventions in effect over the course of the UK epidemic, however, to support interpretation of the forecast performance, we also characterised them based on the trajectory of the epidemic within each period as detailed in Table 2.

### Age-specific transmission parameters

Finally, to compare the implicit assumptions within the models we applied, we assessed how the values of the relative susceptibility and infectiousness parameters s and i varied over the pandemic. To provide an interpretable quantification of these parameters, we used the age-specific values estimated in the model to calculate ratios of susceptibility and infectiousness of younger adults, and older adults relative to that of children. Due to the different age-stratification in the data available, the broader age-bands here varied between case forecast models and infection forecast models: Children were defined as up to 15 for infections and up to 19 for cases, younger adults were defined as 16–49 for infections and 20–49 for cases and older adults were defined as over 50 in both instances.

## Results

### Forecasts

We made forecasts with a horizon of one, two, three and four weeks at fortnightly intervals (Figs 2 and B—G in S1 Text) between 30th October 2020 and 26th November 2021 (29 forecast dates). Visually the forecasts deviate more from the true data at longer forecast horizons.

### Model evaluation

To assess the relative performance of each of the models for different forecast horizons, we calculated evaluation metrics separately for each forecast horizon across all forecast dates (Fig 3 A and Table A in S1 Text). For an alternative approach using multivariate evaluation across age groups and time horizons, see Held et al. (2017) [25].

When forecasting case reports from the UK Covid-19 dashboard [29], the non-interaction model performed better than any of the models which allowed interaction. The next best model was the model with no contact data which performed similarly to the no-interaction model, particularly at longer time horizons with rCRPS of 1.02, 0.99, 1.01 and 1.01 (relative to the no-interaction model) respectively at horizons of one to four weeks in ascending order. The two models that incorporated contact data both performed poorly when considering the CRPS relative to the no-interaction model, with the POLYMOD model performing the worst. However, the relative performance of both models improved at longer time horizons, with the CoMix model performing similarly to the other NGM models at four week horizons (rCRPS of 1.53 and 3.78 at one week horizon reducing to 1.01 and 1.53 at four week horizon for the CoMix and POLYMOD data models respectively). Both CoMix and POLYMOD forecasts had a substantial positive bias, showing that, on average, they over-predicted cases with bias between 0.15 and 0.25. The other models tended to under predict cases by a smaller margin (between 0 and 0.18). The exponential baseline performed worse than the no-interaction model at all forecast horizons (rCRPS 1.35, 1.25, 1.11 and 1.09 at one to four weeks in ascending order). The fixed value baseline initially performed second to worst for one week forecasts (rCRPS = 1.76) but improved as the horizon increased, eventually becoming the best performing forecast at four week horizons (rCRPS = 0.74).

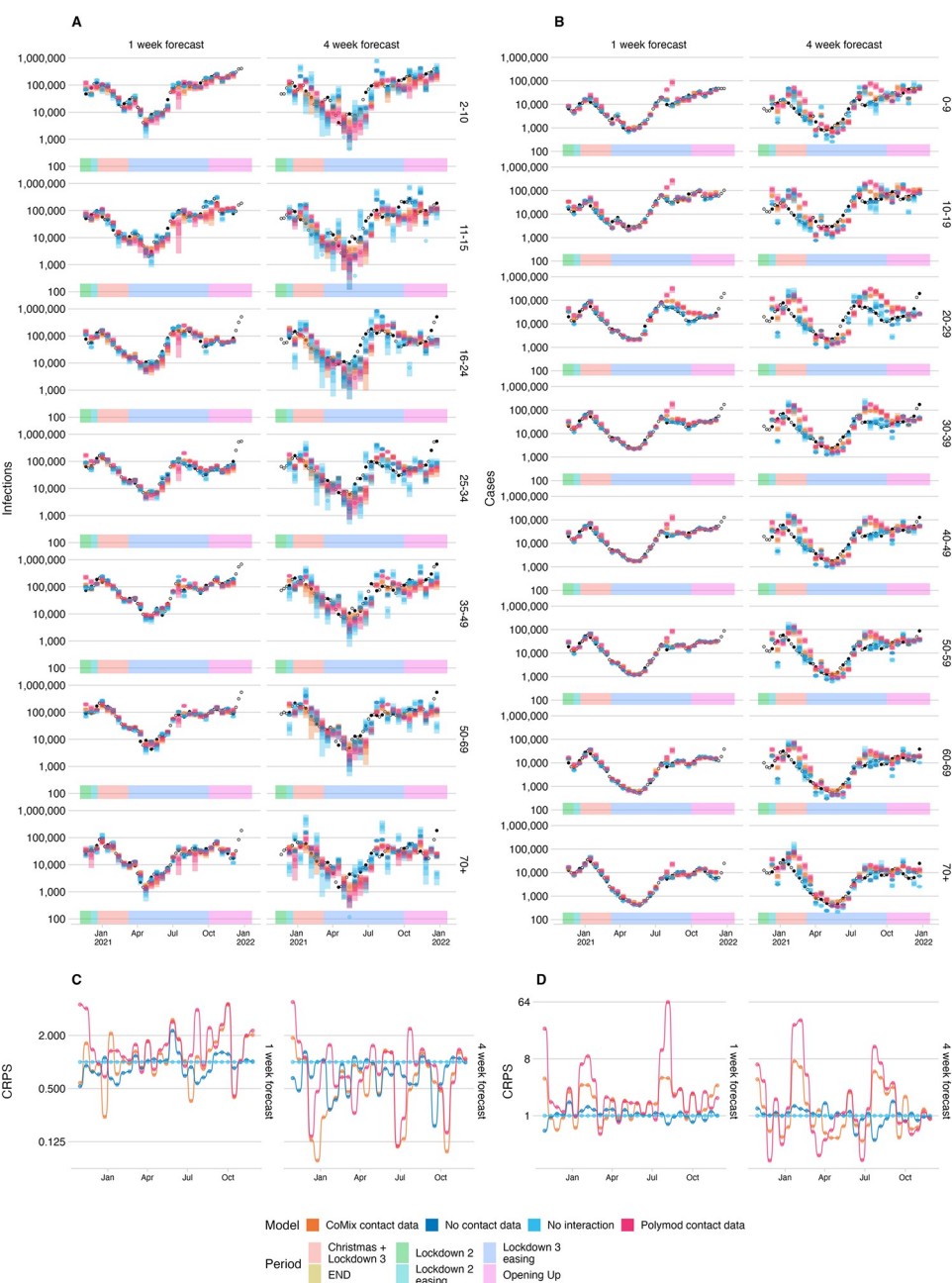

**Fig 2.** A) and B) Infections and cases, respectively, forecast using the CoMix-data based next generation model, the no-contact-data, no-interaction and POLYMOD-data data based model, for each age group (top to bottom) and forecast horizon (left to right). projected infections from each model (coloured bars) and black points show infection estimates and reported cases in plots A and B respectively. The estimates being forecast on each axis are shown as solid points; those not being forecast are shown as rings. C) and D) show the continuous ranked probability score relative to the score of the "no interaction" model for each forecast date, calculated from the Infection and Case forecasts respectively.

When forecasting infection incidence estimated from UK prevalence survey data [27], the no-interaction model performed second only to the no-contact-data model (rCRPS = 0.89) for horizons of one week, followed closely by the CoMix-data model (rCRPS = 1.05). The POLY-MOD-data model performed worst when forecasting one week horizons with a rCRPS of 1.21.

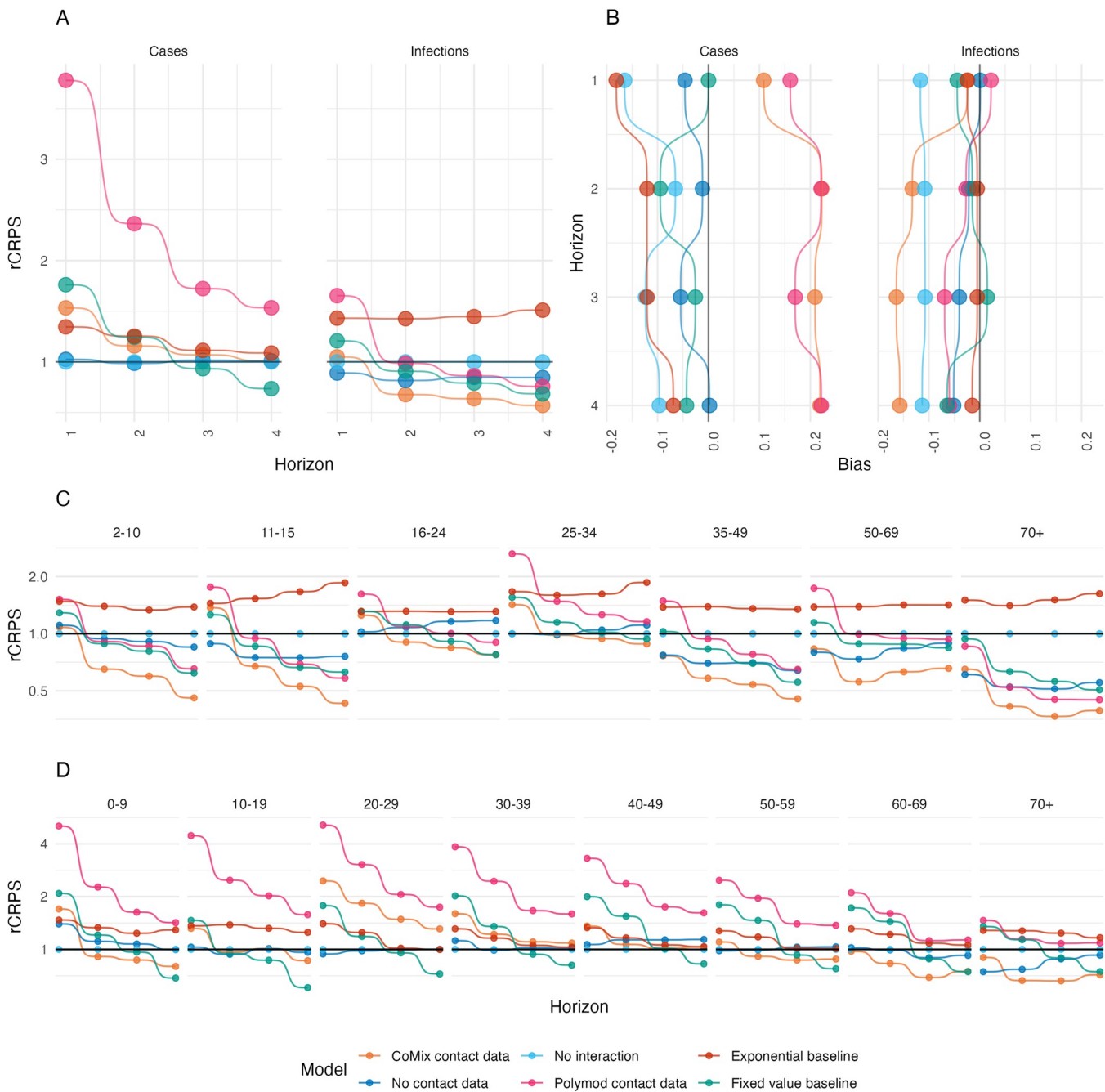

**Fig 3. Continuous ranked probability score relative to the score of the no-interaction model.** A and B show overall rCRPS (A) and Bias (B) of each model when applied to case (left) and infection (right) data for each forecast horizon. B. and C. show the CRPS relative to the no-interaction model against forecast horizon disaggregated by age for infection and case data respectively. The colour of the points shows the corresponding model.

However, at two week horizons the no-interaction model became the worst performing model overall—which remained true for three and four week forecast horizons. In these cases the CoMix-model performed best of all the models including the baseline models with rCRPS of 0.68, 0.64 and 0.57 (relative to the no-interaction model) for two, three and four week horizons respectively. The second best performing NGM model at two and three week horizons was the

no-contact-data model, rCRPS of 0.82, 0.85 respectively. At four week horizons the POLY-MOD-data model was second best performing NGM model with a rCRPS of 0.76. The base-lines both did worse than all but the POLYMOD-data model when forecasting at a one week horizon, however the performance of the fixed value baseline improved relative to all of the NGM models at longer forecast horizons and produced the second best performing forecasts overall for forecast horizons of three and four weeks (after the CoMix-data model) with rCRPS of 0.79 and 0.68 respectively.

We compared the relative forecast performance scoring predictions in each age group separately (Figs 3, B, and C in S1 Text and Table B in S1 Text). When forecasting infection incidence, we found that the CoMix model and no-contact-data model forecast infections better than the no-interaction model in middle-aged adults and older adults (35+ years old) for all forecast horizons. The models also performed best at forecast horizons of two weeks or more in young children (2–10 years old) and older adults. In contrast, the no-interaction model performed much more similarly to the interaction models for forecasts within younger adults (16–34 years old). The same was also true for older age groups (60+) in the case forecasts but not for children, middle aged adults or children. For infections, the performance of all the models improved in all age categories relative to the exponential extrapolation baseline as forecast horizon increased, the fixed value baseline improved relative to the no-interaction model in all age categories but provided poorer forecasts than the CoMix model in all age-categories and time horizons. For case forecasts however the fixed value baseline improved relative to all of the models as horizon increased, providing the best forecasts at four week horizons in age groups between 0 and 59 years.

We divided the analysis into seven periods (Fig 4 and Table C in S1 Text), within each of which national restrictions on social activity remained broadly consistent. For consistency we used the same periods as those presented in Gimma et. al. [19], which presents the key findings of the CoMix study. The relative improvement in performance for the CoMix-data model was most consistent when forecasting infections during the *Christmas and Lockdown 3* period, which was the only period where the CoMix-data model performed the best overall at all forecast horizons. When forecasting cases, the CoMix-data model also performed relatively well during this period at forecast horizons of two or more weeks, but performed comparably to the no-interaction model at one week forecast horizon. The Comix model's infection forecasts outperformed all other NGM models in the two periods following this (*Lockdown 3 easing* and *Opening up*) for forecast horizons of two weeks or more, where only the fixed value baseline model improved on its score. Similarly for the *Lockdown 2 easing* period, the CoMix-data model performed better than the no-interaction model at all forecast horizons, but the no-contact-data model performed better for one and two week forecast horizons. The improved performance of the CoMix model was not wholly reflected in the case forecasts. In particular, the CoMix model performed more poorly than the no-interaction model at all time horizons during the *Lockdown 3 easing* period. However the CoMix model performed better during the *Lockdown 2* period, than the other NGM models for case forecasts, whereas for infection forecasts the CoMix model performed comparably to the no-interaction model during this period.

## Forecast calibration

Of the four NGM models, the CoMix data-based model was best calibrated for case forecasts at one and two week horizons and at all horizons for infection forecasts. This is evidenced by closer agreement between the proportion of true values in each central range of the predictive distribution (50% and 90%) with the value of the range (Fig 5A and 5B). Calibration typically

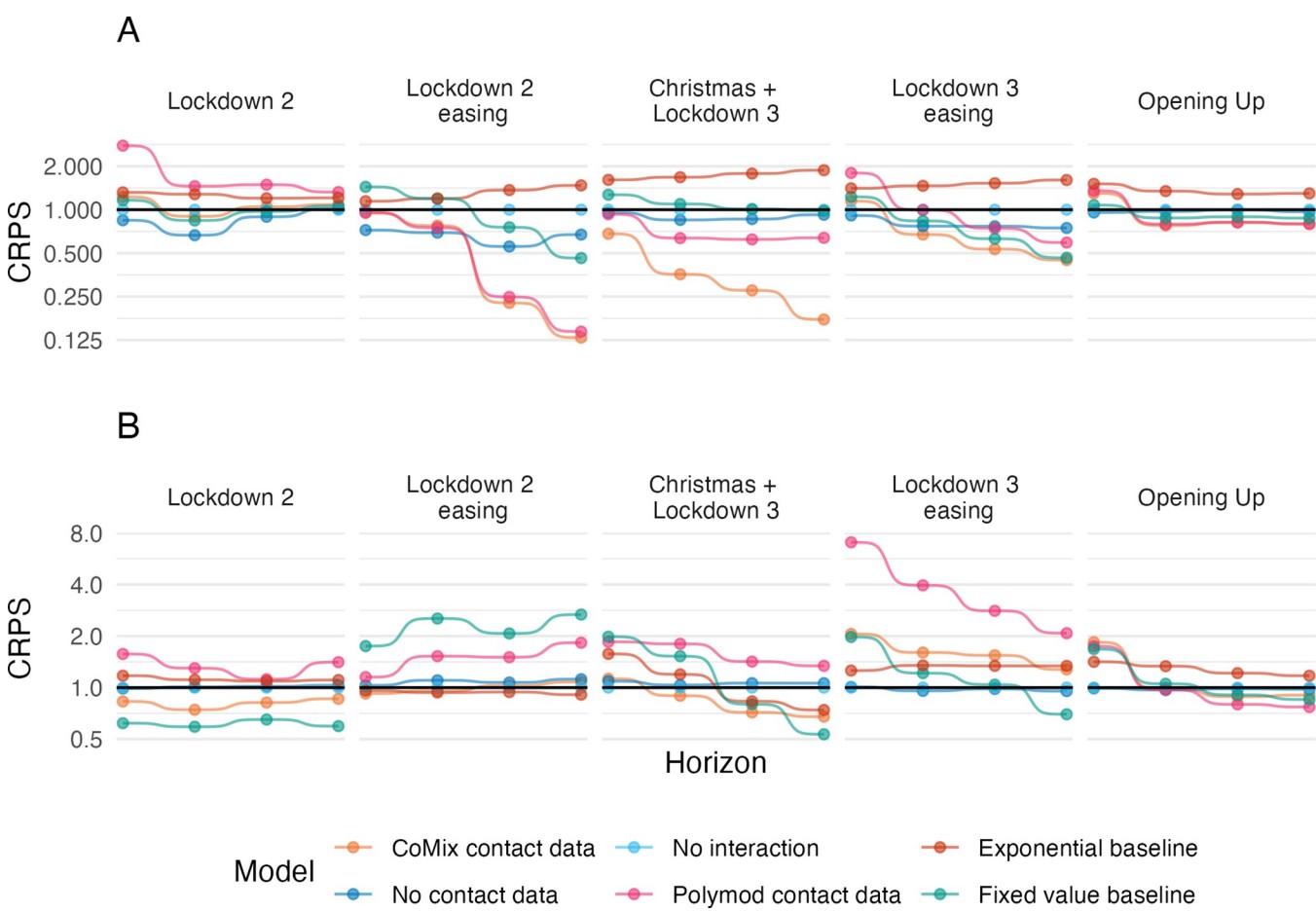

**Fig 4. Continuous ranked probability score relative to the score of the no-interaction model against increasing forecast horizon (one to four weeks).** Panels left to right show each pandemic period, Panels top to bottom show forecasts of cases and infections.

became poorer at longer forecast horizons, with more true values falling outside the specified ranges than would be expected. We also note that none of the forecasts were particularly well calibrated when considered overall forecast dates. The best performing forecast, infection forecasts made by the CoMix model at a one week horizon, saw fewer than 75% of true values fall within the 90% confidence range of the associated projections and fewer than 40% within the 75% confidence range. Separating the forecasts by period of the pandemic (Figs H and I in S1 Text) revealed that the CoMix model was best calibrated for 'Christmas and Lockdown 3' and 'Lockdown 3' periods, for both the case and infection forecasts. In particular the CoMix model's forecast of infections was very well calibrated during the 'Christmas and Lockdown 3' period, with more than 80% of true values falling within the 90% confidence range of the forecast at all horizons. The other models were also relatively well calibrated during these periods. Overall the other periods were much more poorly calibrated. In particular, the "Lockdown 2" and "Lockdown 2 easing" periods were very poorly calibrated across all models with no true values falling within the 90% confidence range for the no-contact-data and no-interaction model forecasts during the 'Lockdown 2' period. The baseline models are not presented as they do not include confidence ranges—hence the forecast coverage is zero for all forecasts by definition.

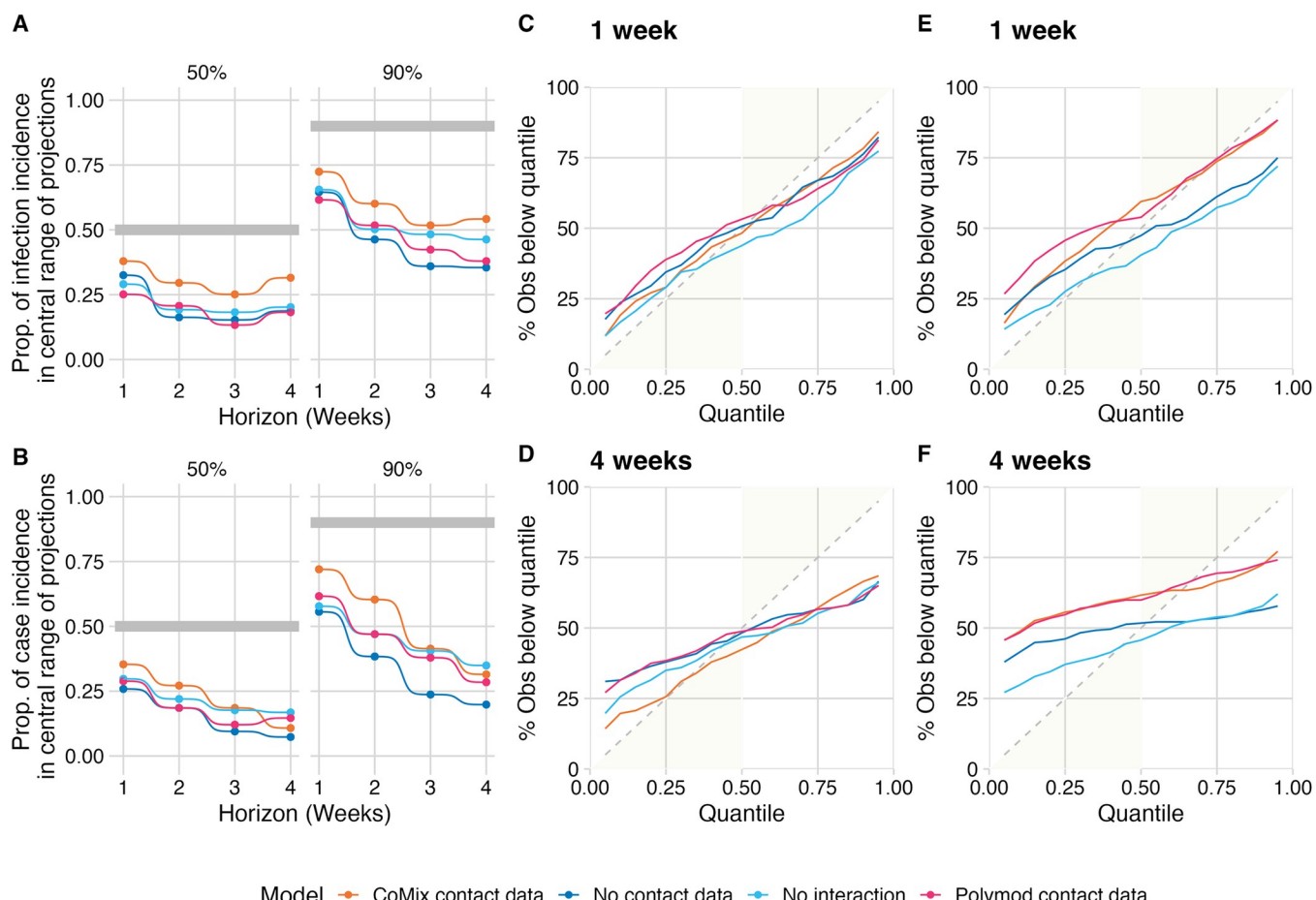

**Fig 5. Calibration of the forecasts made by each of the next-generation-matrix-based models.** A) The proportion of observed mean incidence of infection estimates (obtained using *inc2prev*) and B) the proportion of observed case numbers that fall within the 50% and 90% central interval of the relevant forecasts of the four models. C) and D) the percentage of observed mean incidence of infection estimates (obtained using *inc2prev*) falling below each quantile of the forecasts at one and four week horizons respectively. E) and F) the percentage of observed case counts falling below each quantile of the forecasts at one and four week horizons respectively.

## Age-specific susceptibility and infectiousness

We extracted posterior estimates of time-varying age-specific infectiousness (i) and susceptibility (s) to assess the biological plausibility of these parameters (Fig 6). For the CoMix model the susceptibility of younger adults (16–49 years old for infections and 20–49 for cases) and older adults (50+ years old) was higher relative to children (under 16 for infections and under 20 for cases). In the early part of 2021, this began to shift such that first susceptibility reduced relative to children in the older adults and then in younger adults. Ultimately by the end of the period evaluated, children had higher susceptibility relative to all adults. A similar pattern was present in all models that allowed interaction between age groups. Infectiousness broadly remained equal between age groups, with the exception of a small number of outlying values within which there is no clear trend.

## Discussion

Evaluating the forecast performance of four next generation matrix models, we found that allowing interaction between age groups and using regularly collected contact data did not

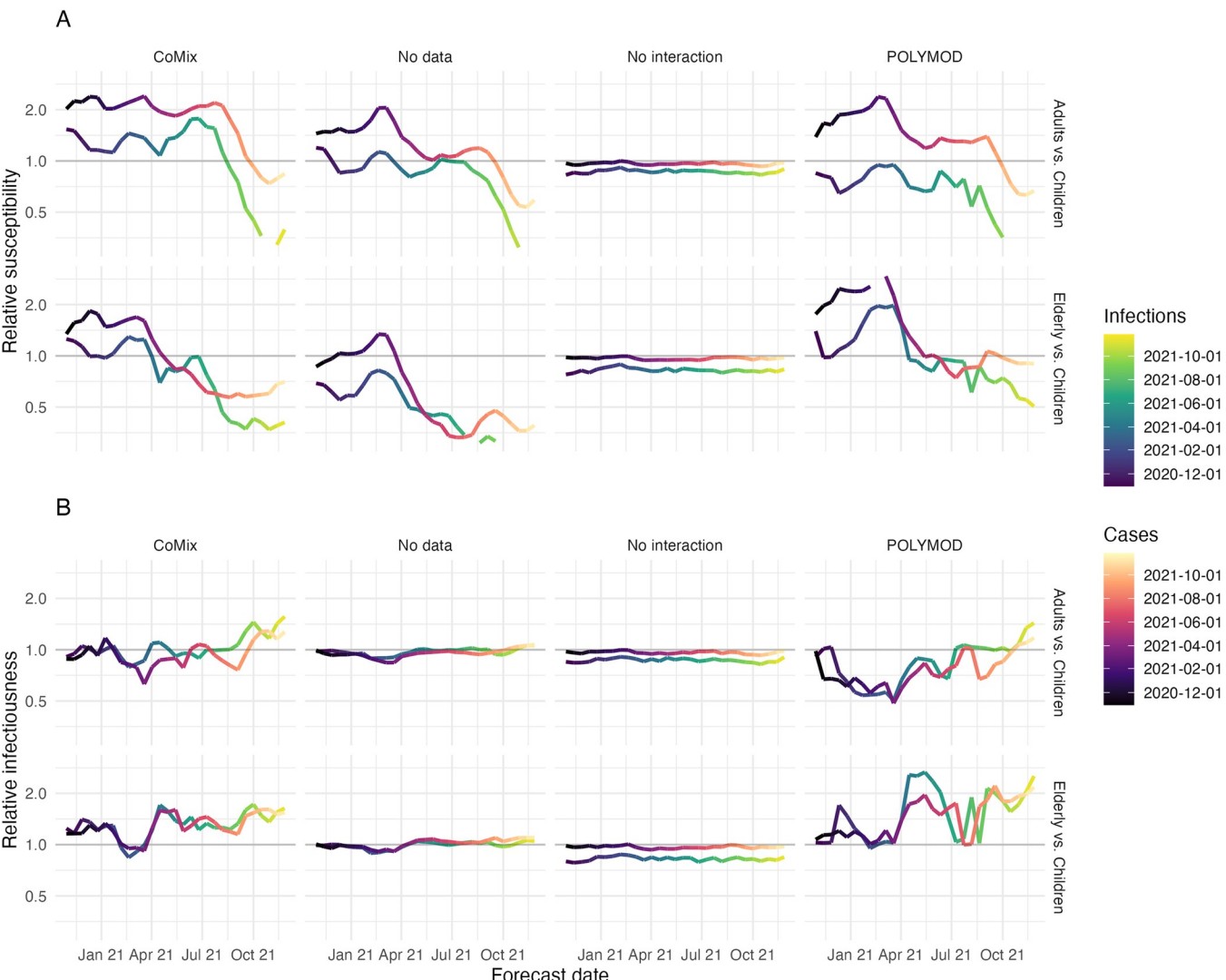

**Fig 6. Susceptibility (A) and** Infectiousness (B) and of younger adults (16–50 for infections and 20–50 for cases) and that of elderly adults (50+) each relative to children (under 16 for infections and under 20 for cases) by forecast date (hue). Each model is shown in panels left to right.

consistently improve performance when forecasting cases as reported through the UK COVID-19 dashboard. However, when forecasting infection incidence estimated from prevalence survey data, forecast performance was improved overall by allowing interaction between age groups at all time horizons. We found that informing interaction by regularly collected contact data improved forecasts further at time horizons of two weeks and greater. Although we found that the improvement was not consistent across all periods or when considering the resulting forecasts for each age group separately.

The NGM models with interaction showed the most benefit over the no-interaction model when forecasting infections during the *Christmas and Lockdown 3* period. Here the CoMix-data model outperformed all other models. During this period the CoMix-data model also proved to be the best calibrated of any model during any period. It's notable that the forecasts were made at a time where the most intense restrictions were imposed for an extended period of time following a very sharp rise in cases. The sharp rise in cases combined with growing

hospitalisations and deaths may have resulted in a period of consistent behaviour amongst the population, since although restrictions changed on January 5th [41] the contact behaviour recorded by CoMix remained similar between *Christmas* and *Lockdown 3 [19]*. This consistency of behaviour, well described by CoMix data, over an extended period of time may ultimately support the performance of this model over the others. Christmas and Lockdown 3 is also the only period of sustained decline in infections, this may indicate that the CoMix model performs best during the declining phase of an epidemic. However, this isn't supported by the forecast performance in the lockdown 3 easing period, where the performance of the CoMix informed model was best when infections began to increase again.

Generally, the NGM models that allowed interaction between age groups performed better than the model with no interaction for infection forecasts. This was particularly true when considering performance in older and younger age groups. This effect may relate to the age-specific incidence and transmission rates. Whereas infections in the younger adults groups were largely driven by transmission within the age group, for long periods of the pandemic infections in elderly and children are likely to have been driven primarily by transmission from the younger adults age groups, particularly when schools were closed, which was the case for a large proportion of the studied period [7,42]—making incidence projections in these groups more reliant on between-age group interaction.

Overall, all forecasts performed better than the exponential extrapolation baseline when forecasting infections. The relative performance of this baseline compared to all other models generally worsened over time suggesting that the simplistic exponential growth model tends to overestimate any change in infections over time, which is compounded at longer horizons. Although the relative performance of the simple exponential extrapolation was better for case forecasts than infection forecasts at short time horizons, similarly to the infection forecasts, all models improved relative to this baseline as the forecast horizon increased, mostly surpassing it to provide better predictions at a longer horizon, showing that this simple assumption of transmission dynamics breaks down rapidly.

In contrast, in both case and infection forecasts, the relative performance of the fixed value baseline improved with increased forecast horizon as all of the modelled values deviated from the true values over time in the case of all models. This may represent the rapidly changing behaviour of the public under constantly changing interventions. This however, also compounds existing evidence that effective forecasts of infectious disease incidence can rarely be made at horizons of greater than a few weeks [6].

The distributions of relative infectiousness and susceptibility inferred by the models are consistent with others findings, beginning with adults exhibiting higher susceptibility than children in general [7,8,16]. This changes throughout the pandemic, following a sequence consistent with what may be expected as a result of acquisition of immunity through vaccination and natural infection. The largest changes occur after vaccination is introduced, where the susceptibility of the older adults reduces relative to other ages first, followed by susceptibility of younger adults. This is consistent with the vaccine roll out schedule in England during the early part of 2021 [43]. The general trajectory of age-specific susceptibility also agrees well with findings of Franco et. al. [43,44] which used the Belgian arm of the CoMix study to estimate age-specific infectiousness and susceptibility independently to this study.

Our estimates of age-specific infectiousness and susceptibility need to be interpreted with caution for three main reasons. Firstly, the framework is optimised for prediction as opposed to inference and therefore is not set up to best reflect the biological processes at play but rather to make good predictions. Secondly, there is likely to be some bias in the way contact data is collected by age which may impact these estimates [19]. Importantly, contacts of children are reported by their parents or guardians. In addition, children's contacts are disproportionately

reported as groups—markedly different from adult contacts, which were reported by the participant themselves and were mostly reported as individual contacts. Finally, we also make no differentiation between contacts by location, duration or intimacy. In reality contacts made in different contexts (e.g. home and school) are likely to carry different potential of transmission, which may also affect the way our susceptibility and infectiousness estimates can be interpreted. One potential extension would be to include contacts by setting (Work, School, Home and Other), which would allow contacts in different contexts to be weighted differently.

There may also be other factors associated with inferred changes in susceptibility and infectiousness which do not correspond to inherent transmissibility. For example, the degree of mitigating behaviours unrelated to contact rate (e.g., masks, preferring outdoor meetings, physical distancing) may have changed differently over the epidemic for each age group. A reduction in relative susceptibility in older adults may indicate that these age groups were able to reduce the risk of infection even when making contact with others further into the pandemic than younger age groups. Also, we assumed immunity is determined by seropositivity as reported in the publicly available CIS data [27], from which we only used a single antibody level threshold for positivity. It may be the case that there is substantial variation in the antibody level distribution in sero-positive individuals of different age groups based on the distribution of vaccine history and infection acquired antibodies, which may affect age-stratified susceptibility to infection. Finally, there may be variation in the age-profile of susceptibility by variant, however due to the limitations discussed, we are unable to quantify this here.

Whereas the relative performance of the models was fairly consistent for infections, the performance when applied to case data was generally more erratic with the ranking for models and baselines changing between horizons within the same aggregation of forecast dates and age groups. This may reflect the more variable nature of case reporting, which is affected by multiple external factors affecting the week-by-week variation in cases beyond transmission dynamics. Notably, case reports are subject to variation in reporting rate, which may also differ between age groups. This is exacerbated by changes in the UK Government's testing policy over the course of the pandemic. This was not the case with the infection time-series, which was estimated from weekly prevalence estimates. Moreover, the infection forecasts incorporated estimates of antibody prevalence modelled from weekly serosurveys [27] and vaccination data [29], whereas the case forecasts did not.

Our work provides an indication of the potential benefits of including contact data in epidemiological forecasts, but for transparency, in our analysis we have chosen not to use state of the art methods of surveillance, instead there are a number of simplifications we made when selecting and processing the epidemiological data to provide clearer analysis of this effect. These simplifications would be expected to affect the performance of forecasts when implemented in real time. Firstly, in our analysis we forecast infections using a modelled time-series fit to weekly prevalence estimates [26]. In truth, under the current data sharing protocol of the ONS Covid-19 infection survey, this data would not be publicly available on the forecast date and hence is not, in this form, applicable as a real-time application without fully integrating into the ONS infection survey workflow. We chose to do this to provide the most idealised scenario to test the application of contact data to short term forecasts, without the complexities associated with case data. Furthermore, although these estimates agree well with other estimates and case time-series, the methodology promotes a smooth infection history leading to autocorrelation in the time-series. This may unduly benefit models with high autocorrelation properties, e.g., the fixed value baseline. However, the similar relative performance of this model when evaluating case data supports our observations that this model performs best at longer time horizons. Secondly, an important feature of real-time epidemiological data is that there are several complex delay distributions which may affect the recent time series of cases

[45]. This is especially true when using data by the date at which individuals are swabbed as we do here, where full information of cases at specimen date are not available until all tests from that date have been processed. For this reason, case counts are increasingly truncated in the days leading up to the forecast date. Approaches to account for this exist [28], but here we have used the case data as known now as opposed to as known on each forecast date, as such we did not need to make this adjustment—as we would if we were making the forecasts in real-time. Extending existing approaches for real-time modelling that can deal with truncated data to include interactions between multiple time series will be an important area of future research [46,47]. In addition, there are many alternative model specifications that could have been incorporated into our analysis. In the interest of clarity, we chose to include only NGM based models, emphasising our evaluation of the use of contact data and not model specification.

The models we present used a normal likelihood, unconventional for epidemiological forecasts which tend to operate on count data. In our case, we use estimates of infection incidence, our input data is therefore not an integer time series, but a distribution at each time-point. To keep the estimates consistent between the case and infection time series' we maintained this approach for forecasting cases as well. Lastly, the absolute measure provided by CRPS means that the overall score is weighted towards age groups and time periods where the absolute incidence was high, this may negatively impact the overall score of models which did poorly in the "young adults" age range (16–35) where incidence was highest for much of the study period.

Overall, allowing interaction between age groups and integrating regularly collected contact data improved forecasts when forecasting infections based on estimates from national prevalence surveys. This benefit was, however, not clear when applied to regularly collected case data, which is generally much more readily available for real-time applications. The picture this offers of the usefulness of contacts in forecasts is nuanced. Even for the idealised example of incident infections estimated retrospectively from repeated cross-sectional prevalence surveys, there are periods of improved performance, and times where the contact-based models failed to capture the dynamics of the epidemic sufficiently to improve on the other models' predictions. The period where the contact data performed the best was during a period where contacts remained relatively consistent. This raises the question as to whether real-time contact data is capable of capturing relevant changes in transmission related behaviour when implementation of non-pharmaceutical interventions are regularly changed. As applications using contact data in real-time develop, it is important to evaluate whether the periods where contact data are informative are aligned with periods when they are also useful for infection control, and consider how future studies might be optimised to ensure this target can be achieved. In summary, our analysis broadly supports the collection of both infection prevalence study data and contact data and their use in short-term epidemic forecasts, but also highlights the limitations of this data. We acknowledge that these studies are resource intensive and we therefore advocate for further in-depth evaluation of contact survey design to ensure they capture changes in behaviour that closely reflects changes in transmission.

## Supporting information

**S1 Text. Supplementary information for evaluating the use of social contact data to produce age-specific forecasts of SARS-CoV-2 incidence.**
(DOCX)

**S1 Data. Excel spreadsheet containing, in separate sheets, the underlying numerical data and statistical analysis for the figures in the main manuscript.**
(XLSX)

## Acknowledgments

The authors would like to acknowledge the contributions of colleagues from the COVID-19 infection Survey Analysis team at the Office for National Statistics (ONS) for their project support and thoughtful discussion during the planning and analysis phase of this research. Also, the members of the CoMix consortium for their support with the contact data, especially Chris Jarvis, Pietro Colletti, Niel Hens and John Edmunds for their feedback on the manuscript. Thirdly, Lloyd Chapman for insightful discussion during the analysis phase of the work. Finally, other members of the Epiforecasts group at LSHTM for helpful comments and feedback on our modelling framework, especially Nikos Bosse for support with the *scoringutils* package.

## Author Contributions

**Conceptualization:** James D. Munday, Sebastian Funk.

**Data curation:** James D. Munday, Sebastian Funk.

**Formal analysis:** James D. Munday.

**Funding acquisition:** James D. Munday, Sebastian Funk.

**Investigation:** James D. Munday.

**Methodology:** James D. Munday, Sebastian Funk.

**Project administration:** James D. Munday.

**Software:** James D. Munday.

**Supervision:** Sebastian Funk.

**Validation:** James D. Munday.

**Visualization:** James D. Munday.

**Writing – original draft:** James D. Munday.

**Writing – review & editing:** James D. Munday, Sam Abbott, Sophie Meakin, Sebastian Funk.

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
