## [Decision Letter · Decision Letter 0]

9 Feb 2023

Dear Dr Munday,

Thank you very much for submitting your manuscript "Evaluating the use of social contact data to produce age-specific forecasts of SARS-CoV-2 incidence" for consideration at PLOS Computational Biology.

As with all papers reviewed by the journal, your manuscript was reviewed by members of the editorial board and by several independent reviewers. In light of the reviews (below this email), we would like to invite the resubmission of a significantly-revised version that takes into account the reviewers' comments.

We cannot make any decision about publication until we have seen the revised manuscript and your response to the reviewers' comments. Your revised manuscript is also likely to be sent to reviewers for further evaluation.

Sincerely,

Claudio José Struchiner, M.D., Sc.D.

Academic Editor

PLOS Computational Biology

Thomas Leitner

Section Editor

PLOS Computational Biology

Reviewer's Responses to Questions

**Comments to the Authors:**

Reviewer #1: This is an interesting manuscript comparing different methodologies for conducting short-term forecasts of COVID-19 data in England.

Major comments:

My main concern is that the manuscript is very descriptive (e.g. the authors show us which method seems to provide better forecasts during which time periods), but the manuscript lacks insights and a discussion why this could be the case.

1) This could be addressed in describing a little more what the major differences are between the Comix and Polymod matrices. Did the authors attempt to use the "physical only contact" matrices from Polymod? Would they be more similar to the Comix matrices over long time periods ?

2) I was missing also a more indepth analysis of performance as a function whether the outcome statistic is "increasing", "stable" or "decreasing". This is only alluded to in the final paragraph of the discussion. I think it would be useful to integrate this more into the analysis. This might make the paper more interesting for persons wanting to use forecasting for different purposes and different diseases.

3) While the statistical methodology seems appropriately chosen, it is not easy to gauge the "significance" of the observed differences of CRPS between the models. Could the observed differences have arisen by chance (e.g. there are clear sample size differences for the different pandemic periods) ? There are obviously differences in degrees of freedom when fitting data - as Comix matrices vary over time and POLYMOD stay the same. Isn't there a risk of overfitting, which is the most parsimonious approach ? What are the criteria to determine whether a method is better, overall, in light of these differences ?

4) In addition to Fig. 6, it would be useful to also provide graphs of estimated average infectiousness and susceptibility over time (possibly as supplementary information) to support statements like in line 520.

5) I was a missing a discussion of what the findings mean for public health and for pandemic preparedness. Are such forecasts useful and what do the authors suggest should be planned ahead for the next pandemic. Will we need contact surveys and infection prevalence surveys ?

Minor:

- Wrong years in abstract line 23 p.1 (october 2021 and November 2022 ?).

- Figure 1: the right hand side legends indicating the colour for each age group is hard to read. Actually it is also a bit superfluous as the age groups are indicated clearly on top of each graph. I suggest to remove the rhs legend.

- From 2021 onwards, seroprevalence is also influenced by vaccination, so it would be necessary to add a supplementary file with cumulative vaccination coverage per age group to Figure 1.

Reviewer #2: I commend the authors for trying to answer an interesting question in regards to whether including detailed contact information improves forecasting performance for epidemic models -- in particular for COVID-19. My comments are attached in PDF report, which is also accompanied by a marked up PDF version of the submitted manuscript.

Reviewer #3: I read with great interest the manuscript PCOMPBIOL-D-22-01784, "Evaluating the use of social contact data to produce age-specific forecasts of SARS-CoV-2 incidence". The authors propose short-term forecasting COVID-19 cases and SARS-CoV-2 (incidence in the United Kingdom via Gaussian multivariate transmission model. The author proposed a transmission model that uses a next generation matrix to induce in the mean term age-groups interaction and include a transmission interval distribution. The some particular cases of the proposed method are applied to COVID-19 cases available in the UK COVID-19 dashboard and to SARS-CoV-2 infection incidence estimated elsewhere from prevalence data. Their proposed model is interesting and there are some points I would like to discuss to make my understanding more clear and some other points to be corrected/updated.

Title: I think the title should be updated by replacing "forecasts" to "short-term forecasts", and by adding the country name. Since this approach seems to be very specific to UK data.

Data sources: For COVID-19 cases, the data come from the UK COVID-19 dashboard, it is the usual number of COVID cases data, are cases reported on that week or cases whose onset symptoms occur on that week? What was the date associated with a case? The weekly infection incidence is an estimate and it is a function of SARS-CoV-2 prevalence and antibody prevalence collected as part of an infection survey, and since it is a survey there is must be some uncertainty on these estimates. And such uncertainty is ignored in the proposed model.

The proposed transmission models are essentially multivariate Gaussian models with a next generation matrix added in the vector mean (Eqs [Disp-formula pcbi.1011453.e029] and [Disp-formula pcbi.1011453.e033]). The four models differ by the way the matrix N(t) is built ([Disp-formula pcbi.1011453.e005]), in particular how the contact matrix C is built. So:

Model CoMix: C(t) is weekly estimated from the UK arm of the CoMix data

No interaction model: C(t) is a identity matrix for all t.

Polymod model: C(t) = C for all t and C is estimated from a pre-pandemic survey.

No contact data: C(t) ? This one I did not understood how the contact matrix was estimated.

The manuscript notation is not simple to follow, and there are some typos. There is no difference among, scalars, vectors and matrices in the equations, so it is not clear that I(t) in Eq ([Disp-formula pcbi.1011453.e001]) is a vector where each element contains the estimated incidence for each age-group at time t. In the text, it is clear that N(t) and C(t) are matrices but it is not clear in equations 1, 3, 6,and 9.

In equation (3), it is not clear what diag(s) or diag(i) means. I assume they are diagnoal matrices with vectors s and i in that diagonal. Again, it is not a clear notation.

Equation (4), in $s_{ab,a}$ what does b mean?

Equation (5), function $A_{a}(t)$ is not defined. In table 1, it is called antibodies. I assume it is the antibody prevalence for age-group a at time t, and it follows a [0,1] truncated Gaussian distribution with undefined mean and variance. (Table 1)

Priors in table 1. In page 11, line 250, the authors claim to use uninformative priors for C_{aa}, (CV_I, CV_C and \\sigma_{cm}) and \\Phi. They do not seem to be uninformative. I am not sure if a large prior sensitivity analysis would be required here, but it would be interesting to explore for a couple of those parameter a set of different priors with smaller and greater variances in at least one of the proposed models to check how robust is the inference. Antibody protection (Phi) and CV_I or CV_c.

In equation 8, it seems to be a circular problem by using the I_\\mu(t) random vector as a function of its own standard deviation. Perhaps in equation (6) the random vector is I(t) with mean I_\\mu(t) which should be equation (1).

Equation 10, I believe it should be $\\sigma_c(t)$ instead of $\\sigma_I(t)$.

For model evaluation (page 18) the authors use two other "models" as baseline. They are not really models unless it is explicitly said that the probability distribution for those estimates is a zero variance variance distribution centred on the proposed estimates. Why not add a Gaussian noise to make it random, or perhaps use a usual statistical time series data-driven model to do k-step ahead forecast. Then I would suggest an ARIMA-like multivariate model or a Bayesian linear dynamical model as West and Harrison (1997, https://doi.org/10.1007/b98971). The advantage is that a baseline statistical model would be better for a comparison. For instance, I wonder if an s_{max} order autoregressive model would be too different than an NGM transmission model with no interaction. Also the uncertainty for the k-step ahead forecasts would also be dealt with since it is an statistical model after all.

Still on model evaluation (page 18), the usual continuous ranked probability score (CRPS) is a scoring rule for an univariate quantity, so for a given age-group and a k-step ahead forecast the CRPS would be calculated using the posterior predictive distribution of incidence I_a(T+k) where a is the age group, T is index for the last observed week and k are the number weeks ahead to forecast (analogously it works for cases c_a(t)). In this case, the CRPS is clear, they are presented in Figure 3B for instance. I am not sure how the CRPS was calculated for the multivariate version, which are the main likelihoods in the manuscript (equations 6 and 9). Is it a multivariate CRPS or is it a CRPS calculated over a function of the vector leading to a scalar, for example I^*_(T+k) = \\sum_a I_a(T+k) which would be the total incidences forecasted for week T+k. This is important to clarify because I am not sure what I am looking in Figures 2C, 3A and 4. Also in Figures 2,3 and 4, even though the y-label says CRPS it should be rCRPS.

Forecast calibration (page 22). All models underestimate the uncertainty (Figure 5A), and the proposed model behaves in a counter intuitive fashion. I would expect that the credible intervals would increase as we increase the horizon, and therefore we would observe more points inside the intervals until we get an intervals that are so large that any point would in it the proportion of points inside the interval would converge to 100%. That is not what is happening in the proposed models, of course data could lead to strange behaviours this is one of the reasons I would like to see one or two usual statistical time series model as baseline models.

Results and discussions. My comments are conditioned on my understanding on the model (I was guessing what the notation should be, I may be wrong in some guesses), on the CRPS used for the multivariate likelihoods and also on the use of different baseline statistical models for a better comparison.

In Figure 6, I am not sure what is meant by infectiousness and susceptibility here. In page 24, the authors extract these quantities from the "model fits". What does model fits mean? Is this a predictive posterior quantity? Which one? Or is it a function of some estimated parameters?

**Have the authors made all data and (if applicable) computational code underlying the findings in their manuscript fully available?**

Reviewer #1: Yes

Reviewer #2: Yes

Reviewer #3: Yes

PLOS authors have the option to publish the peer review history of their article (what does this mean?). If published, this will include your full peer review and any attached files.

Reviewer #1: No

Reviewer #2: **Yes: **Luiz Max Carvalho

Reviewer #3: No
---

## [Decision Letter · Decision Letter 1]

26 Jul 2023

Dear Dr Munday,

Thank you very much for submitting your manuscript "Evaluating the use of social contact data to produce age-specific short-term forecasts of SARS-CoV-2 incidence in England" for consideration at PLOS Computational Biology. As with all papers reviewed by the journal, your manuscript was reviewed by members of the editorial board and by several independent reviewers. The reviewers appreciated the attention to an important topic. Based on the reviews, we are likely to accept this manuscript for publication, providing that you modify the manuscript according to the review recommendations.

Sincerely,

Claudio José Struchiner, M.D., Sc.D.

Academic Editor

PLOS Computational Biology

Thomas Leitner

Section Editor

PLOS Computational Biology

Reviewer's Responses to Questions

**Comments to the Authors:**

Reviewer #2: I would like to warmly compliment the authors on their thorough revision work. I'm pleased with the changes made to the paper and genuinely think it'll be a much better piece because of them. Unfortunately, I do not have a reference to recommend on the issue of heterogeneity in antibody levels across age-strata, so I suppose we will have to drop the issue for now.

Reviewer #3: The authors have answered all of my questions. The new notation has improved a lot my understanding of the proposed models.

In general, the CoMix model was still the best model for infection incidence from prevalence survey data. However for cases, adding age interactions and contact data did not improve performance according to CRSP. I understand the CRPS is an usual measure to evaluate model performance. The authors correctly pointed that out. But in my experience, a probabilistic model that represents/emulates the complexity of real data would lead to a reduction of predictive uncertainty (captured for instance in a reduction of the range of the prediction intervals), which in my humble opinion would be a better improvement. I am not sure if the CRPS would capture such improvement.

I am not suggesting to try a different predictive measure, that would imply on a chance of the baseline model. That is why I suggested to change it to a simpler probabilistic model like an ARIMA or a state-space model. Again I understand the author argument of using far more basic (near deterministic) models and I am OK with that argument. So I am happy with the manuscript the way it is now.

Typo:

Line 226. After equation 3. veci probably should be \\vec{i}

**Have the authors made all data and (if applicable) computational code underlying the findings in their manuscript fully available?**

Reviewer #2: Yes

Reviewer #3: Yes

PLOS authors have the option to publish the peer review history of their article (what does this mean?). If published, this will include your full peer review and any attached files.

Reviewer #2: **Yes: **Luiz Max Carvalho

Reviewer #3: No

Figure Files:

Data Requirements:

Reproducibility:

References:

---

## [Editor Report · Decision Letter 2]

21 Aug 2023

Dear Dr Munday,

We are pleased to inform you that your manuscript 'Evaluating the use of social contact data to produce age-specific short-term forecasts of SARS-CoV-2 incidence in England' has been provisionally accepted for publication in PLOS Computational Biology.

Best regards,

Claudio José Struchiner, M.D., Sc.D.

Academic Editor

PLOS Computational Biology

Thomas Leitner

Section Editor

PLOS Computational Biology

---

## [Editor Report · Acceptance letter]

7 Sep 2023

PCOMPBIOL-D-22-01784R2 

Evaluating the use of social contact data to produce age-specific short-term forecasts of SARS-CoV-2 incidence in England

Dear Dr Munday,

I am pleased to inform you that your manuscript has been formally accepted for publication in PLOS Computational Biology. Your manuscript is now with our production department and you will be notified of the publication date in due course.

With kind regards,

Jazmin Toth
